# TEER and Ion Selective Transwell-Integrated Sensors System for Caco-2 Cell Model

**DOI:** 10.3390/mi14030496

**Published:** 2023-02-21

**Authors:** Elisa Sciurti, Laura Blasi, Carmela Tania Prontera, Amilcare Barca, Lucia Giampetruzzi, Tiziano Verri, Pietro Aleardo Siciliano, Luca Francioso

**Affiliations:** 1National Research Council of Italy, Institute for Microelectronics and Microsystems, 73100 Lecce, Italy; 2Department of Biological and Environmental Sciences and Technologies (DiSTeBA), University of Salento, 73100 Lecce, Italy

**Keywords:** organ-on-a-chip (OoC), anodic stripping voltammetry, transepithelial electrical resistance (TEER), electrochemical impedance spectroscopy (EIS), Caco-2 monolayer, Transwell™ plate

## Abstract

Monitoring of ions in real-time directly in cell culture systems and in organ-on-a-chip platforms represents a significant investigation tool to understand ion regulation and distribution in the body and ions’ involvement in biological mechanisms and specific pathologies. Innovative flexible sensors coupling electrochemical stripping analysis (square wave anodic stripping voltammetry, SWASV) with an ion selective membrane (ISM) were developed and integrated in Transwell™ cell culture systems to investigate the transport of zinc and copper ions across a human intestinal Caco-2 cell monolayer. The fabricated ion-selective sensors demonstrated good sensitivity (1 × 10^−11^ M ion concentration) and low detection limits, consistent with pathophysiological cellular concentration ranges. A non-invasive electrochemical impedance spectroscopy (EIS) analysis, in situ, across a selected spectrum of frequencies (10–10^5^ Hz), and an equivalent circuit fitting were employed to obtain useful electrical parameters for cellular barrier integrity monitoring. Transepithelial electrical resistance (TEER) data and immunofluorescent images were used to validate the intestinal epithelial integrity and the permeability enhancer effect of ethylene glycol-bis(2-aminoethylether)-N,N,N’,N’-tetraacetic acid (EGTA) treatment. The proposed devices represent a real prospective tool for monitoring cellular and molecular events and for studies on gut metabolism/permeability. They will enable a rapid integration of these sensors into gut-on-chip systems.

## 1. Introduction

From health-care applications to environmental monitoring, the development of sensing platforms has attracted significant research interest. The last decade has shown a growing demand for rapid, low cost and easy-to-use technologies for real-time analysis as screening tools or diagnostic systems [1,2,3]. In this field, innovative bioelectronic devices, integrating biological systems with chemical/physical sensors, can provide the opportunity to monitor different parameters associated with specific diseases, metabolic disorders and physiological alterations [4,5]. Currently, biosensing approaches are increasingly employed as a component of micro-engineered cell culture systems, better known as organ-on-a-chip (OoC) devices. OoC are miniaturized in vitro 2D/3D cell culture platforms able to recapitulate physiological functions, cellular organization and features of different organs and tissues [6,7,8,9].

Among various biomarkers, ion concentration detection in different matrices represents one of the main analytical challenges in biology [10], materials [11], food [12] and environmental [13] sciences. The possibility of measuring the concentration of ions, such as H^+^, Na^+^, Zn^2+^, Cu^2+^, K^+^, Ca^2+^, Mg^2+^, NO_2_^−^ and NO_3_^−^_,_ can be an important instrument in many biological studies since ions usually play a key role in cellular and organism-wide signaling, metabolism, and regulation [10]. For instance, zinc and copper are important trace elements involved in many physiological and biochemical processes in the body. Zinc plays a crucial role in several cellular mechanisms, including intestinal ion transport and the integrity of the gastrointestinal tract; its homeostatic imbalance is associated with several intestinal diseases such as inflammatory bowel disease and colorectal cancer [14,15,16]. Copper is an essential micronutrient, important for a correct cell function, and implicated also in the integrity of the intestinal epithelium [17,18]. Homeostasis of metal ions in the body is crucial, and an imbalance in their concentrations or mis-localization of these ions is associated with specific pathologies such as Menkes, Wilson’s and Alzheimer’s diseases. Understanding is needed of how these metal ions are regulated and distributed in the body [19].

The most common and highly sensitive techniques for the quantitative determination of ions include atomic absorption spectroscopy (AAS), inductively coupled plasma-mass spectroscopy (ICP-MS) and inductively coupled plasma-atomic emission spectroscopy (ICP-AES) [20]. However, although these laboratory-based techniques are characterized by high performance and low detection limits, their equipment is bulky, and the analyses are time-consuming, expensive and often require complex sample pretreatments. In addition, the integration of AAS or ICP detection setups is not feasible in transwell or microfluidic devices.

In the last few years, electrochemical analytical technologies have emerged in the field of ion detection due to rapid measurement, high selectivity, low power consumption and low cost of analysis [21]. Among them, anodic stripping voltammetry (ASV) is a powerful technique for trace analysis of metal ions, such as Cu^2+^, Pb^2+^, Cd^2+^, and Zn^2+^ with low detection limits, good sensitivity, and selectivity [22]. Two main steps characterize ASV: a first pre-concentration step at negative potential for a given time, followed by a voltammetric detection scan. The pre-concentration step enables electroplating of the ion of interest on the working electrode, while during the voltammetric measurement the metal is stripped from the working electrode, giving an oxidation current peak. The main advantage of the described technique relies on the possibility to tune the pre-concentration time in order to detect metal ions at very low concentrations [23].

In contrast, ion selective electrodes (ISEs) are used also for potentiometric detection of heavy metal ions. In a potentiometric ISE, the potential change induced by the accumulation of the charged ion of interest is monitored [24,25]. This kind of electrode exploits an ion-selective membrane (ISM) which contains an ionophore, that reversibly binds a specific type of ion, promoting its detection [26]. However, the spontaneous process of ion accumulation on the electrode, without a pre-concentration step, produces a small potential change, which limits the ISE response causing a poor detection limit (LOD) [27,28].

Coupling electrochemical stripping analysis with the ion selective membrane (ISM), as reported in [27], represents a good strategy to improve the selectivity (due to the ionophore) and the sensitivity (due to the ASV technique) of the sensor. In addition, both the presence of a selective membrane and a pretreatment step of impregnation prior to ASV reduce the possibility of detecting interfering metals [23]. In the present work, copper and zinc selective membranes were engineered and deposited on two separated working electrodes on the same device allowing the transport of the selected ions and their detection using square wave anodic stripping voltammetry [29]. Flexible and miniaturized sensors were fabricated and applied to measure the permeation of copper and zinc in real-time across an intestinal epithelium model.

Caco-2 cell line, isolated from human colorectal adenocarcinoma, is frequently used as a model for the intestinal epithelium to study biological transport phenomenon in vitro [30,31]. Caco-2 cells grow in 21 days in a polarized monolayer characterized by the formation of tight junctions between adjacent cells, separating the apical and the basolateral membrane compartments [32,33,34]. The integrity of the epithelial monolayer is typically assessed in real-time through transepithelial electrical resistance (TEER) measurements, performed in a non-invasive manner via ohmic resistance-based or impedance-based measurements [35,36].

In this study, Caco-2 cells were seeded in Transwell™ systems which are frequently used for transport studies, permeability investigations and as comparative static systems to evaluate the accuracy of the involved organ-on-a-chip model [37,38]. After 21 days, Caco-2 cell monolayers were subjected to impedance spectroscopy analysis, in situ, across a selected spectrum of frequencies (10–10^5^ Hz); an equivalent circuit was applied to fit impedance spectrum data to obtain electrical parameters as indicators of the cellular barrier integrity. Ethylene glycol-bis(2-aminoethylether)-N,N,N’,N’-tetraacetic acid (EGTA), a calcium chelator, was used to induce controlled tight-loss of epithelial cell tight junctions resulting in an increased permeability of the Caco-2 monolayer [39]. Flexible polymeric membrane ion selective electrodes were employed to detect zinc and copper ion concentrations in the apical and basolateral chamber of transwell supports, before and after EGTA treatment, supported by TEER measurements and immunofluorescence staining of cells. The experimental set-up and the use of ion selective sensors for cell culture monitoring in situ represent an innovative application of this technology and a first step towards its integration in a novel gut-on-a-chip system.

## 2. Materials and Methods

### 2.1. Materials

Polyvinyl chloride (PVC) of high molecular weight, 2-nitrophenyl octyl ether (o-NPOE), sodium tetrakis [3,5-bis (trifluoromethyl)phenyl] borate (NaTFPB), the copper(II) ionophore I—o-xylylene bis diisobutyldithiocarbamate (o-XBDiBDTC), potassium tetrakis(p-chlorophenyl) borate (KTCPB), the zinc ionophore I—tetra butylthiuram disulfide (TBTDS), tetrahydrofuran (THF), copper chloride, zinc chloride, phosphate buffered saline (PBS, liquid, sterile-filtered, suitable for cell culture), ethylene glycol-bis(2-aminoethylether)-N,N,N’,N’-tetraacetic acid (EGTA), were purchased from Sigma Aldrich, Merck KGaA, Darmstadt, Germany. Bovine Serum Albumin (BSA), E-cadherin/CD324 Recombinant Rabbit Monoclonal Antibody, Alexa Fluor^®^ 647 Goat anti-Rabbit IgG secondary antibody, SlowFade^®^ Gold Antifade Mountant with DAPI were purchased from Thermo Fisher Scientific Inc., Waltham, MA, USA.

### 2.2. Cell Culture and EGTA Treatment

Caco-2 cells (ATCC^®^ HTB-37™, Manassas, VA, USA) were seeded in T-75 cm^2^ flasks and maintained at 37 °C in a humidified atmosphere containing 5% CO_2_ and 5% air. The culture medium adopted was the complete Minimum Essential Eagle Medium (MEM; Corning, NY, USA) supplemented with 20% fetal serum bovine, 1% non-essential amino acids, 1% L-glutamine, 1% penicillin-streptomycin solution. The cells were trypsinized when the confluence reached 80–90% and then they were seeded at 5 × 10^5^ cells cm^−2^ in transwell support suitable for a 6-well plate. A stock solution of 5mM EGTA in PBS, pH adjusted to 7.4 with 1 M Tris-HCl, was prepared and added in Transwell™ cell culture systems for 2 h, to disrupt the Caco-2 monolayer, increasing paracellular permeability.

### 2.3. Sensors Fabrication

Cu^2+^ and Zn^2+^ ion selective electrodes (ISE) were prepared by a viscosity-controlled drop casting of two specific ion selective membranes (ISMs) on different gold working electrodes on flexible Kapton substrate sensors. The electrodes were fabricated on flexible Kapton substrate with an industrial standard process, 50 µm thick, with copper interconnection and pads, 35 µm thick. A standard ENIG (electroless nickel immersion gold) process enabled a 3 µm nickel layer/50 nm gold layer to be been realized onto electrodes and pads areas. The flexible sensor consists of a reference electrode (middle electrode) and two working electrodes (side). The embedded Ag/AgCl reference electrode was prepared by high vacuum evaporation of an Ag layer (99.99% purity), 400 nm thick, onto the middle electrode by shadow masking. The Ag surface was electrochemically chlorinated to AgCl in 0.1 M KCl solution at 0.03 mA/mm^2^ constant current density for 60 s, applied by a Keithley 2450 SourceMeter (Figure 1).

### 2.4. Membrane Blend Fabrication

The zinc ion selective membrane was prepared as reported by Kojima et al. [40]. Briefly, 37.5 mg of PVC were dissolved in 1.25 mL of THF, together with 50 mg (48 µL) of the plasticizer o-NPOE, 5 mg of the zinc ionophore I, and 0.75 mg of the salt with lipophilic anion KTCPB. After the dissolution of all components, the homogeneous mixture in THF was dropped on the working electrode and allowed to evaporate for approximately 24 h at room temperature. The composition of the copper ion selective membrane was based on the studies of Eylem et al. [11], but the relative molar abundances were balanced to that of the zinc membrane, in order to obtain a solution with compatible viscosity and a final membrane with good sensing capabilities. As a final customized protocol, 37.5 mg of PVC was dissolved in 1.25 mL of THF, together with 50 mg (48 µL) of the plasticizer o-NPOE, 6.15 mg of the copper ionophore I, 1.33 mg of the salt with lipophilic anion NaTFPB. After the dissolution of all components, the homogeneous mixture in THF was dropped on the working electrode and allowed to evaporate for approximately 24 h at room temperature before measurement.

### 2.5. TEER Measurements

Impedance spectroscopy of Caco-2 monolayer was performed to define transepithelial electrical resistance (TEER), before and after EGTA treatments. Figure 2a shows the adopted technical solution for a reliable TEER measurement integrated into multiwell platform; two circular transparent electrodes, about 20 mm diameter, were realized with an ITO-coated PET foil and inserted at the bottom of the well and into the upper side of the transwell insert. In order to guarantee a reliable electrical contact, two commercial FPC-to-DIP adapters were used for ITO electrodes [41]. Impedance data were recorded over a frequency range from 10 Hz to 100 kHz, at an amplitude of 10 mV, no bias, using an impedance analyzer (MFIA, Zurich Instruments, Zurich, Switzerland). Impedance spectra were analyzed and fitted with an optimized equivalent electric circuit using Zview software (Scribner Associates Inc., Southern Pines, NC, USA).

### 2.6. Measurements of Ion Sensing Capabilities

A series of Cu(II) and Zn(II) solutions, at various concentrations (1 × 10^−11^, 1 × 10^−10^, 1 × 10^−9^, 1 × 10^−7^, 1 × 10^−6^ M) were prepared in PBS (pH 6.5) starting from copper chloride (CuCl_2_) and zinc chloride (ZnCl_2_) respectively. Electrochemical measurements were performed with Ivium Vertex One potentiostat (Ivium Technologies, Eindhoven, Netherlands) connected to a computer with an electrochemistry software (PS Trace and Ivium). A two-electrode configuration, consisting of a Cu^2+^-ISE and a Zn^2+^-ISE used separately as working electrodes and an integrated Ag/AgCl reference electrode, was applied. Square wave anodic stripping voltammetry (SWASV) tests were performed to calibrate the sensors at different Cu^2+^ concentrations. First, after 5 min of impregnation in the analyte solution, a pre-concentration step at constant −1.0 V for 30 s was conducted in order to accumulate and reduce copper ions on the ion selective membrane. This first step was immediately followed by square wave voltammetry (SWV) from −0.2 to +0.5 V with 4 mV potential step increment, 20 mV pulse amplitude, and 25 Hz frequency; values of the oxidation peak current were plotted as a function of Cu^2+^ concentration, to obtain the calibration graph. The Zn^2+^ calibration plot was performed by SWASV with a pre-concentration step at −1.2 V for 30 s, and the SWV from −1.0 to −0.2 V with 8 mV potential step increment, 50 mV pulse amplitude, and 50 Hz frequency. After every SWASV, a cleaning potential at 0.3 V for 90 s was applied to strip ions from the electrode surface. Transwell™ inserts, employed in cell cultures, were integrated with two different flexible sensors, one for each transwell chamber (apical and basolateral) (Figure 2b). SWASV was carried out for copper and zinc ion detection directly in the Transwell™ cell culture system, with the same method used for the calibration plot, including the intermediate steps of cleaning.

### 2.7. Immunofluorescent Staining

Immunofluorescence analyses were performed on Transwell™ inserts, 21 days after seeding, before and after EGTA-treatment. Caco-2 cells were rinsed with PBS and fixed with 4% paraformaldehyde for 15 min, at room temperature. Followed by blocking step with PBS containing 5% BSA for 1 h at room temperature, the cells were labeled with E-cadherin/CD324 Recombinant Rabbit Monoclonal Antibody at a dilution of 1:500 in PBS with 1% BSA and incubated overnight at 4 °C. After three washes, cells were incubated with Alexa Fluor^®^ 647 Goat anti-Rabbit IgG secondary antibody at a dilution of 1:400 in PBS for 30 min, at room temperature. Lastly, nuclei were stained with SlowFade^®^ Gold Antifade Mountant with DAPI and imaged with Zeiss Axio Imager M2 (Carl Zeiss Microscopy, LLC, White Plains, NY, USA).

## 3. Results and Discussion

### 3.1. Experimental Design

The idea of the present work is to develop an integrable electrochemical sensor able to detect in real-time low concentrations of metal ions of interest in a cell culture system. Experiments of ion sensing were performed in Transwell™ plates inclusive of permeable membrane inserts where Caco-2 cells were seeded to form a polarized epithelial monolayer. Cell differentiation and the consequent formation of intercellular tight junctions, structurally resemble the intestinal epithelium and provide a physico-chemical barrier to the passage of ions and small molecules [31]. Due to these properties, this epithelial model is widely used for ion and drug transport studies. Typically, ion solutions or drug candidates are added to the apical compartment and detection is usually achieved by sampling both apical and basolateral media, to study the epithelium permeability or the distribution of specific ions [19,42]. In this work, flexible sensors for zinc and copper ion detection in the apical side of epithelium (top transwell cavity) and basolateral side of epithelium (bottom transwell cavity) were fabricated and integrated. Moreover, TEER measurements and immunofluorescence staining of cells were performed to validate the permeability of the monolayer before and after EGTA treatments.

### 3.2. TEER Measurements and Equivalent Electrical Circuit Fitting

A conventional method to monitor growth and differentiation of in vitro Caco-2 cell epithelial monolayer is based on Trans-Epithelial Electrical Resistance (TEER) measurements. TEER is a quantitative technique employed to verify the formation of tight junctions in live cells in a quick, non-destructive, and non-invasive manner. In addition, TEER offers the possibility to confirm the integrity of the cellular barrier before studying transepithelial passage of ions, chemicals, or drugs. The standard method for measurement of TEER relies on ohmic resistance calculation as the ratio of the voltage and current [43]. Two electrodes, placed one in the apical compartment and the other in the basolateral side, are used to impose a direct current (DC) voltage and measure the resulting current. TEER, calculated according to Ohm’s law, is the sum of medium, filter and cell layer resistances [44]. Although the method described above is commonly used, even if suffering from measurements errors due to electrodes spacing variations, TEER values can be also extracted through electrochemical impedance spectroscopy (EIS) combined with a fitting equivalent circuit. Impedance spectroscopy is a more accurate and advantageous technique than the classical method of TEER measurements. Applying a small amplitude alternating current (AC) signal across a range of frequencies, and measuring the amplitude and phase response of the resulting current, the impedance data can provide additional information about structural and physical properties of the cellular monolayer, avoiding DC voltage damage risk to cell layer [45]. Indeed, impedance experimental data can be analyzed using an equivalent circuit model, by combining circuit elements with electrical parameters of the cell barrier. Impedance spectroscopy of Caco-2 cells grown on Transwell™ filters for 21 days was recorded, in order to verify the completed formation of the epithelial barrier. Moreover, the impedance response of the cellular monolayer to EGTA treatment, a calcium chelator used to disrupt the epithelial barrier, thus increasing the paracellular permeability and transepithelial resistance, was measured. A stock solution of 5 mM EGTA in PBS was prepared and added to the apical side of the Transwell™ inserts. An impedance analyzer, connected with two electrodes separated in the apical and in the basolateral Transwell™ compartments, was set with an amplitude of 10 mV and at a frequency range from 10 Hz to 100 kHz. The data were fitted with the equivalent circuit depicted in Figure 3a. A resistance element, representing TEER and the ionic flux, and a capacitance element, indicating the capacitive behavior of the cellular membrane, were connected in parallel. The choice to consider two RC-elements in series arises from the assumption of a polarized cell layer with different structure and composition of the cell membrane in the apical and basolateral sides [44]. An additional resistance was considered as the resistive behavior of the medium. Nyquist plots of EIS showed a different response before and after exposure of the Caco-2 layer to EGTA (Figure 3b). When the epithelial barrier was treated with EGTA, the Nyquist graph displayed an altered trend and two semicircles appeared, indicating changes in resistance of apical and basolateral membranes. Fitting parameters confirmed the effect of EGTA on epithelial permeability. The total electrical impedance for the equivalent circuit in Figure 3a, can be calculated by the equation:(1)$$Z=R_{media\;}+\frac{R_{1}}{1+j\omega R_{1}C_{1}}+\frac{R_{2}}{1+j\omega R_{2}C_{2}}$$
where $R_{media\;}$ is the resistance of the medium, *R*_1_ and *C*_1_, *R*_2_ and *C*_2_ are the resistance and the capacitance of the apical and basal side of the epithelium, respectively, *j* is the imaginary unit, and ω is the radial frequency [46]. TEER and capacitance values of the total epithelial layer can be summarized as follows:TEER = R_1_+R_2_
(2)
(3)$$\frac{1}{C}=\frac{1}{C_{1}}\;\;\frac{1}{C_{2}}\;$$

Fitting results showed decreasing values of TEER after 2 h of treatment with EGTA, starting from a value of 746 ± 54 Ω‧cm^2^ to 288 ± 29 Ω‧cm^2^. These data confirmed the effect of the calcium chelator on the integrity of the Caco-2 layer, leading to an increased permeability of the cellular barrier to ions and molecules. In contrast with TEER, the calculated membrane capacitance of the cells has not changed significantly, with an average value of 1.8 ± 0.4 μF/cm^2^, suggesting that EGTA exposure affects mainly tight junctions, without considerably altering the cell membrane structure [47].

### 3.3. Ion-Sensing

In order to guarantee a straightforward integration into Transwell™ cell culture systems, different flexible ion selective electrodes were employed to monitor zinc and copper ions in real-time. The developed ion sensors are based on a gold electrode deposited on Kapton and covered by a polymeric ion selective membrane, employed for anodic stripping voltammetry analysis. The working principle of our sensors is reported in Figure 4. Briefly, the electrode is soaked into each analytical solution to allow the impregnation of the analyte ion into the membrane and to induce the formation of the ion-ionophore complex (Figure 4a, step 1); after the impregnation phase, a negative potential is applied to the working electrode to induce the migration of the ion towards the gold interface and its reduction (electroplating) (Figure 4b, step 2). Finally, a voltammetric scan is performed to “strip” the analyte from the gold surface and to oxidize it, resulting in a current peak proportional to its concentration (Figure 4c, step 3). The reactions of the three steps for both copper and zinc ions are shown below:$$\begin{array}{ll} Cu_{\left(l\right)}^{2+}+L_{\left(ISM\right)}\to Cu^{2+}\;L_{\left(ISM\right)} & (\mathrm{step}\;1);\\ Cu^{2+}\;L_{\left(ISM\right)}\to Cu_{\left(g\right)}^{0} & (\mathrm{step}\;2);\\ Cu_{\left(g\right)}^{0}\to \;Cu^{2+}\;L_{\left(ISM\right)} & (\mathrm{step}\;3). \end{array}$$
$$\begin{array}{ll} Zn_{\left(l\right)}^{2+}+L_{\left(ISM\right)}\to Zn^{2+}\;L_{\left(ISM\right)} & (\mathrm{step}\;1);\\ Zn^{2+}\;L_{\left(ISM\right)}\to Zn_{\left(g\right)}^{0} & (\mathrm{step}\;2);\\ Zn_{\left(g\right)}^{0}\to \;Zn^{2+}\;L_{\left(ISM\right)} & (\mathrm{step}\;3); \end{array}$$
where *L* is the ionophore, *l* the analyte solution, *ISM* the polymeric ion selective membrane, *g* the surface of the gold electrode.

Electrochemical characterization of sensors were performed at various concentrations of Cu(II) chloride (CuCl_2_) and Zn(II) chloride (ZnCl_2_) solutions in PBS (Figure 5a,c). The resulting calibration curves, plotted by the Cu^2+^ and Zn^2+^ oxidation peak current as a function of copper and zinc ions concentrations, were shown in Figure 5b,d. A linear relation was observed for both Cu^2+^ and Zn^2+^ concentrations in the range from 1 × 10^−11^ to 1 × 10^−6^ M (R > 0.99). The peak current increased with the CuCl_2_ and ZnCl_2_ concentrations and the calibration equation was y = 1.64 × 10^−7^ x + 1.97 × 10^−6^ for copper, and y = 5.36 × 10^−7^ x + 6.07 × 10^−6^ for zinc. The relative standard deviation (RSD) for every concentration point was lower than 10.63% for copper and RSD < 8% for zinc. The limit of detection (LOD), calculated using the relation 3 σ/m, where σ is the standard deviation of the peak current for the blank sample and m is the slope of the calibration curve, is approximately 2.51 × 10^−12^ for copper and 6.02 × 10^−12^ for zinc.

After integrating ion selective sensors in a Transwell™ cell culture system, square wave anodic stripping voltammetry (SWASV) was performed to detect zinc and copper ions flux from the apical to the basolateral compartments before and after EGTA treatment, in order to test the ion sensing in real-time. A solution of CuCl_2_ (1 × 10^−6^ M) in PBS (pH 6.5) replacing culture medium was added to the apical side of the transwell in which Caco-2 cells were grown for 21 days, whereas in the bottom side, the medium was replaced with pure PBS. Voltammetry measurements in the first minutes confirmed the highest current peak in the apical side at potential of about 0.1 V (T0, Figure 6a) and the absence of peak for the basolateral side (T0; Figure 6b). When the Caco-2 layer was damaged by EGTA exposure, the copper solution flowed through the damaged junctions, from the apical to the basolateral side. One hour and two hours after the treatment, SWASV measurements were acquired and the resulting current peaks revealed a progressive decrease of copper ions concentration at the top, matching increased current peaks at the bottom side (blue and red lines, Figure 6a,b). The data showed a reduction of about 40% and 70% of the peak current after 1 h and 2 h, respectively, from the EGTA treatment. Similarly to copper detection, zinc chloride (ZnCl_2_) solution (1 × 10^−6^ M) in PBS (pH 6.5) was added in the transwell and the voltammograms at different time points were observed. As shown in Figure 6c,d, an analogous trend was observed also for zinc detection. Current peaks at potential between −0.5 V and −0.4 V were recorded: current peaks decreased by 45% and 54% were measured in the apical side of the transwell 1 h and 2 h after the EGTA treatment, respectively; at the same time increasing concentrations of zinc ions were collected in the basolateral side before and after the disruption of the epithelium.

### 3.4. Immunofluorescent Images

Immunofluorescent staining was performed on the transwell insert, 21 days after seeding, before and after EGTA-exposure, in order to confirm the results obtained from TEER analysis and ion sensing measurements. EGTA acts as a calcium chelator, causing the dissociation and disruption of tight junctions (TJ) and adherent proteins (AJ), usually involved in cell–cell adhesions of the intestinal epithelium. Fluorescence imaging of E-cadherin, a type of adherent protein, is indicative of the Caco-2 monolayer integrity. A confluent cell layer, with marked E-cadherin, was observed before any treatment (Figure 7a). In contrast, after 2 h of EGTA-exposure, fluorescence imaging revealed an altered localization of the adherent proteins, indicating that the Caco-2 layer was compromised (Figure 7b). A reduced distribution of E-cadherin suggests a loss of cell adherence and, as consequence, an increased permeability of the epithelium. These results confirm the decreased values of TEER after EGTA exposure and justify the high passage of ions through the Caco-2 monolayer, from the apical to the basolateral side of the transwell plate.

## 4. Conclusions

The development of a sensors-integrated cell culture platform, for TEER and ion detection measurements in situ and in real-time was reported. Flexible and miniaturized ion selective sensors were fabricated by using different ion selective membranes which enhance selectivity and were employed for SWASV measurements of low ion concentrations in Transwell™ cell culture systems. Electrochemical measurements were performed at different time points in order to evaluate the flux of ions, like zinc and copper, across a Caco-2 monolayer in real-time. Interesting results were obtained from square-wave anodic stripping voltammograms: a clear decrease of ion concentrations was revealed in the apical chamber of transwell after Caco-2 layer disruption through EGTA treatment. In situ impedance spectroscopy and immunofluorescent staining analysis confirmed the effect of the calcium chelator on tight junctions and paracellular permeability of the intestinal monolayer, validating the data obtained from ion sensing measurements. The proposed device represents an innovative tool for monitoring cellular and molecular events, and for studies on gut metabolism and permeability. It sets the stage for chemical (ions) and physical (TEER) sensors integration in the future gut-on-chip microphysiological systems.

## Figures and Tables

**Figure 1 micromachines-14-00496-f001:**
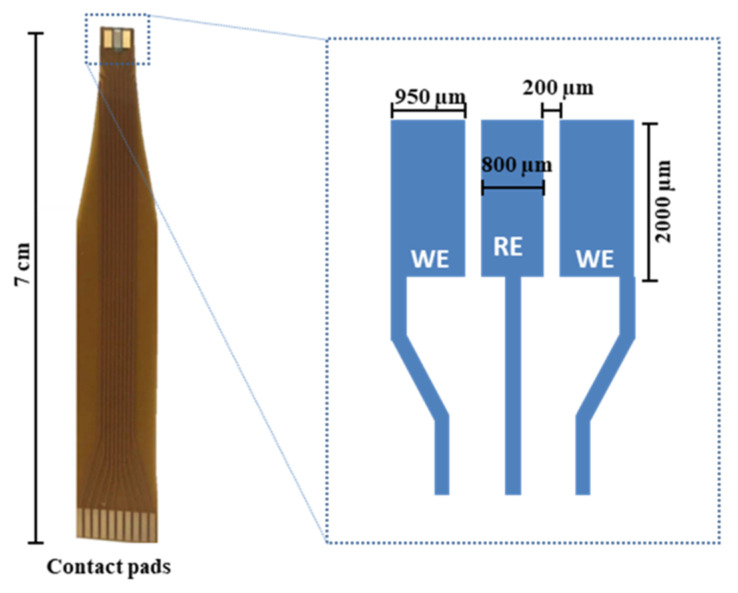
Flexible ion selective electrodes. Photograph of the ion selective sensor on flexible Kapton substrate (70 × 4 mm^2^ tapered tip) and close-up schematic image of the sensitive area structure of sensor. Two different working electrodes supported by a common Ag/AgCl reference allow detection of two different ions.

**Figure 2 micromachines-14-00496-f002:**
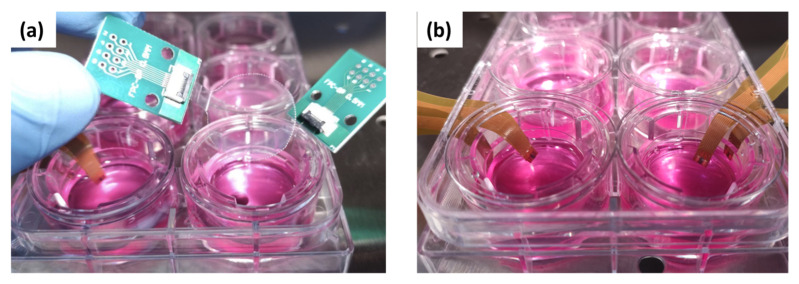
Photographs of sensor-integrated Transwell™ plates. TEER measurement set-up by circular transparent electrodes with FPC-to-DIP adapters (**a**). Ion-selective flexible sensors inserted in transwell plate, one for each compartment (top and bottom) of the well; each sensor can measure two different ions with a common thin film Ag/AgCl reference electrode (black electrode on flex between the two working electrodes) (**b**).

**Figure 3 micromachines-14-00496-f003:**
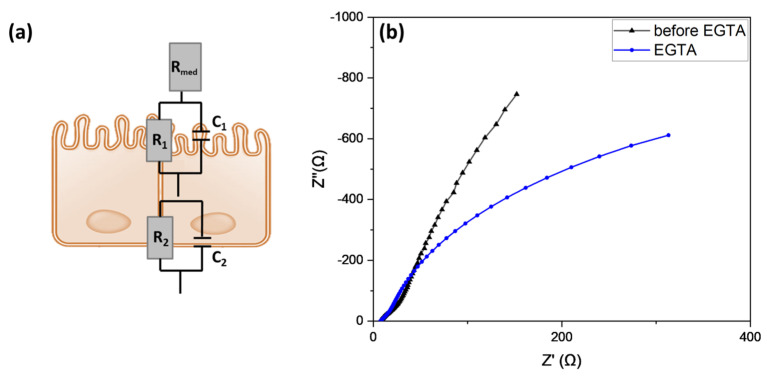
Impedance spectroscopy of 21 days cultivation of Caco-2 cells in transwell inserts, before and after EGTA treatment. Schematic representation of the applied equivalent circuit indicating the electrical parameters of the cell layer (**a**). Nyquist plots of EIS analysis of Caco-2 monolayer between 10 Hz and 100 kHz, before and after 2 h of EGTA exposure (**b**).

**Figure 4 micromachines-14-00496-f004:**
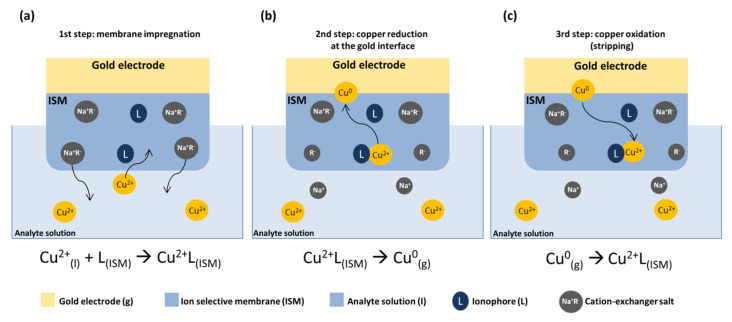
Illustration of the working mechanism of the polymeric ion selective sensors exposed at the established electrochemical protocol: membrane impregnation (**a**); pre-concentration step (**b**); anodic stripping voltammetry (**c**).

**Figure 5 micromachines-14-00496-f005:**
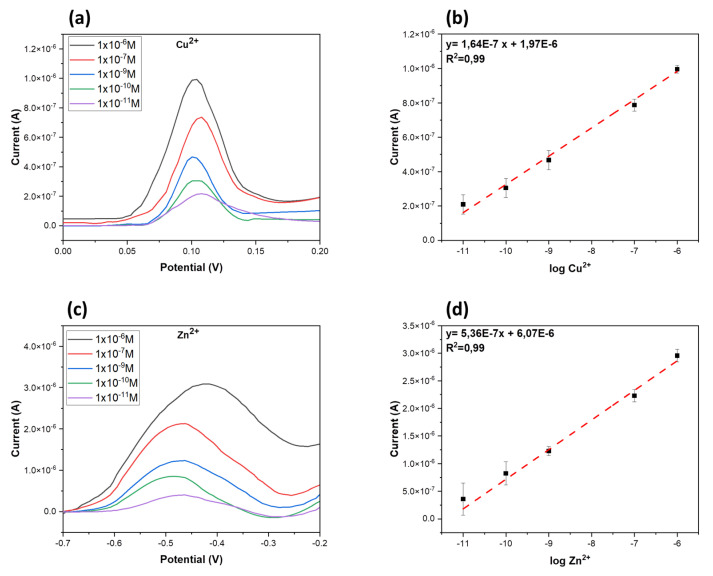
Electrochemical characterization of ion sensors. Square wave voltammograms of Cu^2+^ at different concentrations (1 × 10^−11^, 1 × 10^−10^, 1 × 10^−9^, 1 × 10^−7^, 1 × 10^−6^ M) in PBS (**a**); plot of CuCl_2_ peak currents vs. log Cu^2+^ concentrations (**b**). Square wave voltammograms of Zn^2+^ at different concentrations (1 × 10^−11^, 1 × 10^−10^, 1 × 10^−9^, 1 × 10^−7^, 1 × 10^−6^ M) in PBS (**c**); plot of ZnCl_2_ peak currents vs. log Zn^2+^ concentrations (**d**).

**Figure 6 micromachines-14-00496-f006:**
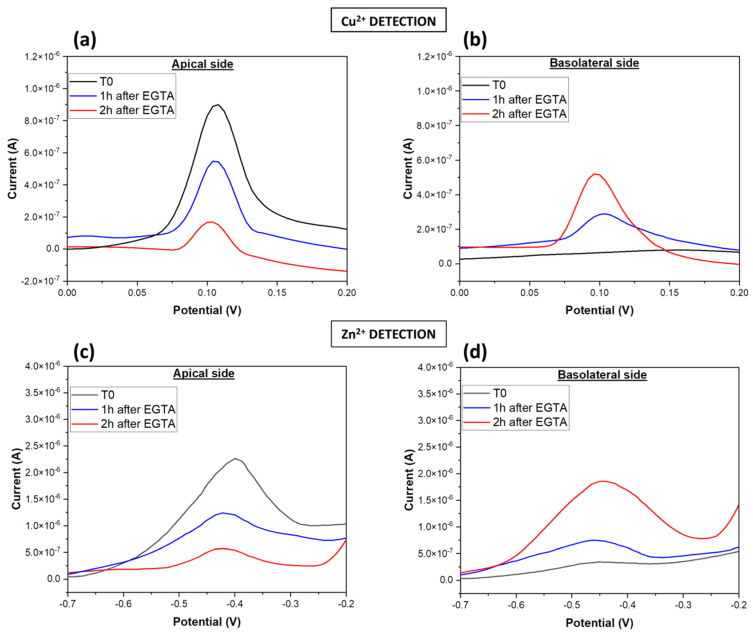
Square-wave anodic stripping voltammograms of Cu^2+^ and Zn^2+^ in Transwell™ cell culture systems. SWASV for copper detection in apical compartment (**a**) and basolateral side (**b**) of transwell Caco-2 culture system at different time points before (T0) and after 1 h, 2 h of EGTA treatment. SWASV for zinc detection in apical compartment (**c**) and basolateral side (**d**) of transwell Caco-2 culture system at different time points before and after EGTA exposure.

**Figure 7 micromachines-14-00496-f007:**
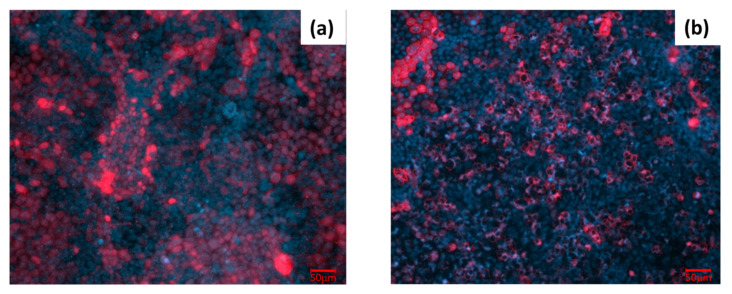
Fluorescent microscopy images of Caco-2 epithelial cell layer. Caco-2 monolayer, 21 days after seeding, before (**a**) and 2 h after (**b**) EGTA exposure. Cells were stained for E-cadherin (anti-E-cadherin Alexa Fluor 647, red) and nuclei (DAPI, blue).

## Data Availability

Not applicable.

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
