# Peer review of "TEER and Ion Selective Transwell-Integrated Sensors System for Caco-2 Cell Model"

_micromachines, 2023, doi:10.3390/mi14030496_

Round 1

Reviewer 1 Report

The authors report an ion sensor which was fabricated with an ion selective membrane for monitoring Zn2+ and Cu2+ ions transport of Caco-2 cell. Compared with previous researches, authors should provide their advantages in detecting ions. In addition, although ion selective membranes could enrich corresponding ions, selectivity of membranes for other ions should be provided in this manuscript. When concentrations of interfere ions are higher than target ions, whether the measured results are influenced by interfere ions.

Author Response

We thank the reviewer for the comments. We focused the work on the possibility to detect metal ions in a cell culture environment through the described electrochemical method. In this respect, we decided to employ commercial ionophores embedded in a membrane with an extensively used composition, innovating that approach and coupling it with ASV technique. The capacity of the reported membranes to selectively bind the target ions was already reported in several papers. On the basis of this evidence, we choose to apply the fabricated sensor in a cell culture system for anodic stripping voltammetry (ASV) measurements of copper and zinc ions. As reported in [1], pretreatments can be performed to reduce the effect of interfering elements. Therefore, we adopted in the paper the membrane impregnation step as gold standard pretreatment to selectively uptake the analyte element into the membrane, thus reducing the effect of other metals in the subsequent ASV steps. We added such concept in the manuscript by adding the following sentence “In addition, both the presence of a selective membrane and a pretreatment step of impregnation prior to ASV reduce la possibility to detect interfering metals”.

  1. Borrill, A.J.; Reily, N.E.; Macpherson, J. V Addressing the Practicalities of Anodic Stripping Voltammetry for Heavy Metal Detection: A Tutorial Review. Analyst 2019, 144, 6834, doi:10.1039/c9an01437c.

Reviewer 2 Report

This is a paper that demonstrates the construction of a membrane and it disruption by the addition of a calcium binder EGTA. The disruption is demonstrated in Figure 3, where the membrane resistance is decreased with the addition of the EGTA.

Two working electrodes were coated with ionophores for Cu and Zn respectively, designed to preconcentrate the metal ions. Next anodic stripping voltammetry was performed. 

My concern about the results is in Figure 5. Typically stripping peak current will increase directly with the metal concentration, rather than the log of the metal concentration (which is more a potentiometric (Nernstian) relationship between the measured potential of an ISE and the log of the metal concentration). However perhaps with the ionophore as a binding agent,  it just happened that within the confines of the experimental method the linear relationship just happened as a result. I am not sure it could be theoretically predicted…

Nonetheless Figure 6 demonstrates the movement of copper and zinc through the disrupted membrane, which supports the results from Figure 3.

The only comment I have is line 14 in section 2.6…’ values of the reduction peak current’ should read ‘vvalues of the oxidation peak current’

Author Response

We thank the reviewer for the positive comments. Regarding the result in Figure 5 we noticed a linear relationship between the stripping peak currents vs. the log of metal concentrations. We agree with the reviewer that this linear relation could be counterintuitive but we attributed these results to the reported “hybrid” electrochemical technique. Indeed, we coupled the electrochemical stripping analysis with an impregnation step by using an ion selective membrane that is usually employed in potentiometric analysis.

We modified the mistake in the line 14, section 2.6.